# Interaction of *Trypanosoma cruzi*, Triatomines and the Microbiota of the Vectors—A Review

**DOI:** 10.3390/microorganisms12050855

**Published:** 2024-04-25

**Authors:** Günter A. Schaub

**Affiliations:** Zoology/Parasitology, Ruhr-University Bochum, Universitätsstr. 150, 44780 Bochum, Germany; guenter.schaub@rub.de

**Keywords:** antibacterial compounds, Chagas disease, interactions, microbiota, mutualistic symbionts, Triatominae, *Trypanosoma cruzi*

## Abstract

This review summarizes the interactions between *Trypanosoma cruzi*, the etiologic agent of Chagas disease, its vectors, triatomines, and the diverse intestinal microbiota of triatomines, which includes mutualistic symbionts, and highlights open questions. *T. cruzi* strains show great biological heterogeneity in their development and their interactions. Triatomines differ from other important vectors of diseases in their ontogeny and the enzymes used to digest blood. Many different bacteria colonize the intestinal tract of triatomines, but only Actinomycetales have been identified as mutualistic symbionts. Effects of the vector on *T. cruzi* are indicated by differences in the ability of *T. cruzi* to establish in the triatomines and in colonization peculiarities, i.e., proliferation mainly in the posterior midgut and rectum and preferential transformation into infectious metacyclic trypomastigotes in the rectum. In addition, certain forms of *T. cruzi* develop after feeding and during starvation of triatomines. Negative effects of *T. cruzi* on the triatomine vectors appear to be particularly evident when the triatomines are stressed and depend on the *T. cruzi* strain. Effects on the intestinal immunity of the triatomines are induced by ingested blood-stage trypomastigotes of *T. cruzi* and affect the populations of many non-symbiotic intestinal bacteria, but not all and not the mutualistic symbionts. After the knockdown of antimicrobial peptides, the number of non-symbiotic bacteria increases and the number of *T. cruzi* decreases. Presumably, in long-term infections, intestinal immunity is suppressed, which supports the growth of specific bacteria, depending on the strain of *T. cruzi*. These interactions may provide an approach to disrupt *T. cruzi* transmission.

## 1. Introduction

The protozoan parasite *Trypanosoma cruzi* [1] (Kinetoplastida, Trypanosomatidae) is the etiologic agent of Chagas disease, also known as American trypanosomiasis [2]. This disease, endemic to Latin America, is also becoming relevant to other countries, especially the United States of America and Canada, but also some European, Eastern Mediterranean, African and Western Pacific countries, as infected people immigrate to these countries [3,4,5]. However, vectorial transmission is limited to the Americas. Only mammals are vertebrate hosts of *T. cruzi* [6].

Since poor people, in particular, are affected, the fight against this neglected disease involves improving housing conditions, as well as insecticide campaigns against the hematophagous insect vectors, the triatomines [7]. Comparing the number of infected people, there was a sharp decline between 1982 and 2010. In 1982, it was estimated that around 20 million people were infected [8]. Due to insecticide campaigns against the domestic vectors, initially in individual countries but now in all Latin American countries, the number fell to 16–18 million people by 1991 and 8 million by 2010 [9,10]. Since then, it has only improved slightly. According to the WHO’s latest estimates in 2018, 6 to 7 million people were still infected worldwide, mostly in Latin America [2]. There, 75 million people are at risk of infection, resulting in 30,000 new cases and 10,000 deaths per year [11]. Even today, 115 years after the disease was first described [1], public health measures against Chagas disease are not organized in many endemic areas [12]. New vector control strategies are proposed based on health education programs and control campaigns, including biological agents, baits, repellents and new insecticides [13].

The parasite is transmitted to humans in various ways, mainly vectorial [13,14]. In this way, feces and/or urine of infected vectors are deposited or smeared on the mucous membranes of the eyes, lips or the wound created by the mouthparts of the triatomines during feeding [14,15]. Allergic reactions to salivary proteins are common in humans, who are recurrent hosts of triatomines [16,17]. This creates scratch wounds that allow further access through the skin. Although around 10,000 infectious metacyclic trypomastigotes are present in the first drop of feces of the triatomine, only around 10 to 1000 invade the experimental host, the mouse, through the skin puncture caused by the mouthparts [18]. Eating infected meat leads to oral infections. Other possibilities are drinks made from sugar cane or fruit juices that contain feces or residues of triatomines. Other routes include organ transplants or blood transfusions from infected people as well as transmission from mother to child [3,14]. 

The course of the disease is divided into two phases, the initial acute phase and the chronic phase [14,19]. In the acute phase, lymph nodes near the site of infection swell and non-specific symptoms such as fever occur. The parasite can often be easily detected in the blood using a light microscope [20]. After about 1–2 months, the disease enters the chronic phase. The indeterminate chronic phase lasts several years to decades and is hardly symptomatic [14,19]. Berenice, in whose blood Dr. Chagas first discovered *T. cruzi* [1], lived in the indeterminate phase for more than 70 years and still had parasites in her blood [19,21,22]. During the chronic phase, parasites are rarely detectable in the blood but can be diagnosed and isolated by hemoculture and xenodiagnoses, and the infection can be diagnosed by serodiagnosis and PCR [20]. The final chronic phase is characterized by organ dysfunction resulting from intracellular development and destruction of cells. The intestinal tract and heart often enlarge into megaorgans. Pathological effects on the heart in particular lead to death [14,23].

There is no vaccination against the parasite. Since 1966 and 1970, respectively, only two compounds, benznidazole and nifurtimox, have been available for treatment and are recommended for the initial acute phase [19,24,25,26]. They often lead to serious side effects and are not effective against all *T. cruzi* strains [27,28,29,30,31]. New drugs are currently being tested, e.g., compounds that interact with the glycosomal membrane transport and inhibitors of trypanosomal topoisomerase II [32,33]. 

In the vector, the triatomine insects, *T. cruzi* colonizes the intestinal tract, but also the excretory system, the Malpighian tubules [34]. In the gut, fungal and bacterial microbiota also develop, including mutualistic symbionts, and all must survive. The interactions between *T. cruzi*, triatomines and the microbiota can be crucial for both the parasites and the bacteria, and studies of parasite and vector interactions are an opportunity to find new ways to interrupt transmission. This review provides an overview of *T. cruzi*, the vectors, the microbiota in triatomines and their interactions.

## 2. *Trypanosoma cruzi*

*T. cruzi* probably originates from a bat trypanosome [19,35]. The trypanosome multiplies by longitudinal divisions, which are rarely preceded by genetic recombination [36], which leads to a predominantly clonal genetic structure of the populations in which panmictic groups appear to occur in addition to purely sexual ones [37,38]. For 12 years, *T. cruzi* strains have been classified by molecular biological methods into six groups of discrete typing units, designated TcI to TcVI [39,40]. Their distribution is only roughly correlated with the different countries (Table 1). Within each typing unit, the strains exhibit great biological heterogeneity (summarized by [31,41,42]). Therefore, it is misleading to consider *T. cruzi* as a homogenous entity and generalize results obtained with a single strain. 

Strong differences are evident in the locations and developmental stages of *T. cruzi* in the mammalian host and in the vectors [46] (Figure 1). In mammals, the parasite invades all types of host cells, except anucleate erythrocytes and thrombocytes, and transforms into amastigotes that multiply intracellularly. Before rupturing the host cell, they develop into nonreplicative stout, slender or intermediate blood trypomastigotes. These enter the blood capillaries and circulate in the blood or infect new host cells. In the vector, the blood trypomastigotes transform into multiplying epimastigotes and spheromastigotes and many different intermediate stages during the development into non-replicative metacyclic trypomastigotes. These differ from epimastigotes by specific changes in the surface coat and the location of the kinetoplast, which is more subterminally located than in blood trypomastigotes [46,47].

## 3. Vectors

Two groups of insects in the order Hemiptera are capable of transmitting *T. cruzi*. They belong to the families Cimicidae (bed bugs) and Reduviidae (assassin bugs) and in the latter to the subfamily Triatominae. While transmission by bed bugs has only been proven in the laboratory [48,49], triatomines have been known as important vectors since the disease was first described [1]. In both families, the piercing–sucking mouthparts are well adapted to ingest blood [7]. About 150 species belong to the Triatominae [50], the largest blood-sucking insects, with females of *Dipetalogaster maxima* reaching 3.8 cm long [51]. Triatominae occur predominantly between the Great Lakes of North America and southern Argentina [52,53]. The majority are sylvatic, but some species are peridomestic, feeding on chickens, guinea pigs, dogs and other hosts around the house. Only some species have adapted to living primarily in houses and are therefore important vectors, e.g., *Triatoma infestans*, *Rhodnius prolixus*, *Panstrongylus megistus* and *Triatoma dimidiata* [7]. None of these major vectors are present in all Latin American countries (Table 1). *R. prolixus* had been widespread in Central America, originating from Venezuela, then escaped from a laboratory in El Salvador and spread north and south until it became eradicated there [45].

### 3.1. Development, Attraction and Blood Ingestion

In these hemimetabolous insects, the development of most species from egg through five nymphal stages to adult takes 5 to 12 months [7,54]. It depends on the ambient temperature, the availability and quality of blood and the supply with mutualistic symbionts. All post-embryonic stages feed on blood, whereas in other important dipteran vectors of disease, only adult insects ingest blood [54]. Under optimal humidity conditions, triatomines can starve for extended periods of time, up to a year [7]. In the field, the nutritional status of populations depends on the season [55,56]. Normally, one full engorgement in each nymphal stage is sufficient for development to the next nymphal or adult stage. If engorgement is disturbed, one or more additional amounts of blood are required [7]. There is an optimal period for feeding. *R. prolixus* nymphs starved for extended periods and exiting shelters in the presence of host odors responded more strongly after 30 than after 21 days of starvation, but the response was not stronger in nymphs starved for 60 days [57]. In the case of the very aggressive triatomine, *D. maxima* [58], the optimal time for feeding is obvious, as the feeding of nymphs within 3 weeks or more than 12 weeks after molt results in more probings and similarly lower amounts of blood ingested than feeding between these times. These nocturnal insects, whose activity is highest at dawn and dusk [59], feed on all terrestrial vertebrates, mainly mammals and birds, but also on warm reptiles and amphibians [60]. Above all, temperature, exhaled carbon dioxide, skin odors and visual stimuli attract the triatomines to the host [59]. There, the proboscis, which covers the thin mouthparts and rests under the body, is pivoted forward, and its tip is pressed against the skin. The mandibles cut the skin. Only the maxillae penetrate and move within the skin until they tap into a blood capillary (summarized by [18]). Saliva is pumped into the blood capillary via a maxillary salivary channel, and the blood is ingested via the large maxillary feeding channel [61]. During a probing phase, the food quality is assessed [62,63]. Blood ingestion is supported by salivary nitric oxide [64], which is bound to lipocalins, the most abundant protein in the saliva (e.g., [65,66,67]). The rate of engorgement is influenced by the host [68]. Depending on their size, triatomines require 3 to 30 min for a full engorgement, 6 to 12 times their own body weight [7].

### 3.2. The Intestine of Triatomines, the Excretory System and the Fate of Blood

The intestinal tube is divided into regions of ectodermal and endodermal origin [69]. Cuticle borders the luminal side of the ectodermal cells, while endodermal cells are lined by the microvilli. After the ectodermal foregut comes the first region of the endodermal midgut, the cardia [7], very often not considered separately. The second region of the midgut is the stomach. [This region is also denominated anterior midgut and crop, but I use the term of Wigglesworth [70].] The third region of the midgut, the small intestine, is divided by a central narrow region into three parts of approximately equal length, the front, middle and back regions [71,72], which are rarely examined separately. The final region is the ectodermal sac-like rectum, also lined by cuticle. 

During blood ingestion, the fastest fluid-secreting cells in the animal kingdom, the upper Malpighian tubuli cells, begin to produce urine [73]. At the end of the four Malpighian tubules are the ampullae, which end at the border between the midgut and hindgut and extend the swollen processes far into the rectal lumen shortly after excretion begins [74,75]. They probably support a special region at the beginning of the rectum, the rectal glands, in absorbing ions, water and other compounds. [Since the designation of the ends of the Malpighian tubules as ampullae is based on the description of Wigglesworth [76], the use of this word for the rectum of triatomines is misleading and must be avoided.] 

The conditions in the intestine of the triatomine are very different from those in the mammalian host [77], where the temperature is about 38 °C, the pH is 7.4, glucose is present as a nutrient for *T. cruzi*, and the immune system contains innate and adaptive components. The temperature in the ectothermic triatomine is usually lower than in the vertebrate host, caused by thermal preferences of vectors in the range of 23 to 30 °C (summarized by [78]). The temperature in the stomach rises due to the blood consumed, but then drops quickly and should then be similar in all intestinal regions. The withdrawal of water changes the consistency, resulting in a jelly-like consistency [79]. After feeding on guinea pigs, the blood crystallizes in the stomach [80]. Within eight days after feeding, the intestinal contents become acidified to pH 5.2, followed by an increase to around pH 6.5 [81]. Compared to the stomach, the time course of pH changes in the small intestine is only slightly delayed. The content is liquefied. In both regions, peristaltic movements mix the contents.

In the rectum, the conditions also change again, very significantly in the first days after feeding. The time course of the changes of the pH is influenced by the urine produced after feeding [82]. When measuring the pH value in the deposited drops of excreta, the pH value in the first dark brown drop is pH 5.9 and corresponds to the 20 days after feeding in the back region of the small intestine [81]. Then, the pH in the excreta rises rapidly up to pH 8.4, which is also the case in urine collected 6–24 h after blood ingestion [82]. The pH in urine collected 24–48 h after feeding, and then in daily periods up to 96 h after feeding, decreases to pH 6.2, although with individual variations between pH 5.7 and 8.3. The changes in osmolality are not as strong and follow a different time course. In the first drop the osmolality is about 320 mosmol/kg H_2_O, followed by about 410 mosmol/kg H_2_O in the fourth drop and a similar value in the rectal contents the following day, but about 760 mosmol/kg H_2_O three days after feeding. Sulfate and potassium dominate in the rectal contents of unfed insects and turn mainly into a sodium chloride solution in the first drops of excreta. One day to ten days after feeding, the concentrations of sulfate, potassium, carbonate, chloride, calcium and sodium fluctuate widely [82]. The consistency also changes within the first few days after blood ingestion. After flushing out the rectal contents, the clear urine resembles a saline solution and then becomes cloudy with yellow-white urate spheres that fill the rectal sac until digestive residues reappear [74,75,82]. Nutrients are hardly present in the urine, but later, the concentration of residues from digestion increases.

The ingested blood passes through the foregut and the cardia and is stored in the strongly expandable stomach. Anticoagulants in saliva and the stomach prevent blood from clotting, and the withdrawal of ions and water concentrates the blood cells [83]. In the stomach, carbohydrates and lipids are digested and absorbed [84]. Erythrocytes are lysed. The activity of aminopeptidases in the contents is higher than in the small intestine [85], perhaps digesting leucocytes, plasma proteins and/or membranes of the erythrocytes. According to light and electron microscopy, the functions of the stomach are ion and fluid transport and nutrient storage [86]. 

Digestion of hemoglobin and absorption of nutrients occurs primarily in the small intestine. The anterior half has a higher frequency of digestive cells than the posterior half and the stomach [87]. The breakdown of hemoglobin is indicated by a color change: the red stomach contents turn brown immediately after passage into the small intestine. After a full engorgement of fifth instar nymphs, digestion takes more than a month in many species. The two main proteases cathepsins B and L are found in the small intestine [88] and the activity of cathepsins L and D in the contents is higher than in the stomach [85,89]. These digestive enzymes differ from those of other blood-sucking insects, e.g., mosquitoes, which use trypsins and chymotrypsins [90]. The development of the perimicrovillar membranes follows the same schedule as in the stomach. In starved triatomines, perimicrovillar membranes do not cover the microvilli, but develop within hours after feeding and then degenerate [91]. Thereby, different compartments are present for different digestive enzymes. Aminopeptidases are located between the perimicrovillar membranes [92] which function as the peritrophic membranes in other insects. Finally, the remnants of digestion are stored in the rectum and the nutrients are absorbed before defecation. During or after blood ingestion, urine flushes out the remnants of digestion from the rectum [53]. 

### 3.3. Immune System of Triatomines 

Like all insects, triatomines have an effective immune system (e.g., [53,93,94,95,96]). In the hemocoel, different types of hemocytes attack pathogens that have penetrated through the cuticle. After blood ingestion, hemocytes, fat body and the cells of the small intestine produce, in a systemic reaction, low-molecular mass compounds, such as nitric oxide, which are involved in antibacterial responses [97]. These and other compounds could enter the intestines. Since nitric oxide is also produced in the nervous system as a diffusible neurotransmitter [98], this system must be strictly separated from the hemolymph. The activity of other immunity enzymes, phenoloxidases, is high in the stomach contents and not detectable in the small intestine [99]. These enzymes could produce toxic quinones against microorganisms. 

In addition, triatomines synthesize many antimicrobial peptides. After their recognition as foreign, the immune response is induced through the signalling pathways Toll, IMD (immune deficiency), JAK/STAT (Janus kinase/signalling transducer and activator of transcription), JNK (Jun-N-terminal kinase) and MAPK (mitogen-activated protein kinase) [95,96,100,101,102,103,104,105,106]. The Toll pathway is activated against Gram-positive bacteria and fungi and the IMD pathway responds to Gram-negative and some Gram-positive bacteria. The JAK/STAT signaling pathway is activated by viruses. Noncoding RNAs regulate the immune response during insect-pathogen interactions [107,108]. The specific memory during subsequent infections with bacteria [109,110] has not been investigated with triatomines. Immune memory against one of two strains of *T. cruzi* is described [111], but in one group, a mortality rate of 70% within 20 days after an infectious feeding is surprising. 

According to the molecular biology data and transcriptomes, several genes are expressed that encode antimicrobial compounds of different masses: defensins (4 kDa), pacifastin-like protease inhibitors (4 kDa), short trialysins (6.1 kDa), a single domain (7 kDa) and a double domain (11.5 kDa) of Kazal inhibitors, diptericin (9 kDa), prolixicin (11 kDa), TiAP (12.5 kDa; bacteriostatic), histones (13, 15 kDa), triatox (14.8 kDa), lysozymes (15 kDa), hemolysin-like proteins (16 kDa), attacins (20 kDa), trialysins (22 kDa; only in salivary glands) and annexins (35 kDa) [65,95,101,104,112,113,114,115,116,117,118,119,120,121]. Some of these antimicrobial peptides exist as multiple isozymes, up to eight defensins, eight trialysins and five lysozymes (summarized by [101]). In *Triatoma* (*Meccus*) *pallidipennis*, 12 different genes encode three mature defensins [122].

## 4. The Microbiota of Triatomines 

Although triatomines ingest sterile blood, various bacteria, fungi and viruses are present in the salivary glands, Malpighian tubules and especially in the intestinal tract [123,124,125,126,127,128]. When ingested, the microbiota first come into contact with the antimicrobial peptides in saliva and then with those from the intestinal wall. Thereby, they are killed or establish in the intestine. Established mirobiota can be classified as pathogenic, commensal or mutualistic and act against non-indigenous species [129]. In *Drosophila*, they induce intestinal homeostasis, are responsible for a basal immune response and affect gut morphology, and the amount of reactive oxygen species is regulated by bacterial density [130,131]. Bacteria can confer resistance to pathogenic bacteria by either activating the immune system or producing toxins. Microbiota composition is also regulated by mutualistic and antagonistic interactions of bacteria [132]. In the following, the term symbiont will only be used for the mutualistic symbionts. 

### 4.1. Infection Routes 

Microbes enter the body of triatomines in various ways. While some viruses are transmitted transovarially from mother to offspring (summarized by [133]), bacteria and intestinal fungi use the oral route. Triatomines ingest water and juices [134,135], which are probably contaminated with bacteria. Before molting, the insects swallow air (which probably contains bacteria) to increase their body size and to rupture the old cuticle. The mouthparts come into contact with the skin before and after ingesting blood, also a source of bacteria. We do not know whether triatomine nymphs contact eggshells after hatching, a behavior known from other Hemiptera [136]. The most important behavior is coprophagy, which is necessary to obtain the symbionts. The drops of deposited feces are contaminated by the bacteria and fungi present in the environment. Triatomines cannot detect whether feces contain symbionts, and coprophagy occurs in all nymphal instars [137]. This is indicated by increasing percentages of infections with the coprophagically transmitted homoxenous trypanosomatid *Blastocrithidia triatomae* in groups where the possibility of coprophagy has existed since the first larval instar, even in the presence of uninfected fifth instar nymphs [138]. In addition, these infections only occur after feeding and not during starvation, and only if nymphs can contact liquid rather than dry feces. This appears to contradict other studies: Nymphs of *T. infestans* are not attracted by freshly deposited feces, but rather by volatile components in dry feces within 24 h of feeding [139]. Because nymphs leave shelters to defecate and such shelters are preferred by other nymphs [140], this attraction could be an adaptation to symbiont transfer, but requires more detailed study.

### 4.2. Microbiota of Triatomines

According to previous culture-dependent identifications of the microbiota, many different bacterial species colonize the intestinal tract of triatomines (summarized by [141]). These numbers increased enormously using molecular biological methods. Approximately 500 bacterial species are present in and on the posterior segments of the abdomen of field-derived *Triatoma dimidiata* [142,143]. Since the recent reviews on the microbiota of triatomines [53,126,127], only a few additional publications have appeared, and some publications were previously unknown. These seven publications used molecular biological methods and are discussed below. 

(I) In all 88 macerated nymphs and one male *Rhodnius pallescens* from palms in Panama, the 16S rRNA sequences indicate similar microbiota [144]. Of these, over 90% of gut communities belong to the phyla Proteobacteria, Actinobacteria (including Actinomycetales), Bacterioidetes and Firmicutes. (II) In adults of 12 *Panstrongylus geniculatus*, seven *Psammolestes arthuri*, eight *R. pallescens*, 22 *R. prolixus*, three *Triatoma maculata* and three *T. venosa*, collected in Colombia in domestic, peridomestic and sylvatic habitats, rRNA sequencing identified 27 bacterial phyla and significant differences between triatomine species [145]. In addition, again, over 90% of the gut communities belong to the same four phyla as in *R. pallescens* from Panama. *Rhodococcus* occurs in *P. geniculatus*, *P. arthuri* and *R. prolixus*, but the identification of the *Rhodococcus* species is still lacking. This is also the case in previous studies (summarized by [127]). (III) In 54 adult *Triatoma sanguisuga* from domestic and peridomestic regions in southern Louisiana [146], almost 80% of the microbiota belong to four taxa, one of them being the Actinomycetales. [However, the symbiont of this triatomine species has not yet been identified.] (IV) This is also evident for the hindgut of 74 adult *Triatoma gerstaeckeri* and *T. sanguisuga* from Texas [147]. Many samples contain bacteria of the genus Actinomycetales, but these bacteria are not present in all triatomines. (V) In investigations of *Triatoma rubrofasciata*, the only triatomine species that is distributed worldwide and develops in ports in the tropics and subtropics [7], four wild-caught adults and 45 nymphs reared from eggs in the laboratory in the first generation also possess a high diversity including Actinobacteria, and the presence of some bacteria correlates positively with origin: laboratory or field [148]. (VI) According to a metagenome shotgun sequencing approach using fifth instar nymphs of *R. prolixus* obtained from a laboratory colony, many different bacteria are also present, with Chlamydiae, Actinobacteria, Firmicutes and Gammaproteobacteria being the most common [149]. (VII) Even in such a sequencing approach of 1 to 16 specimens (nymphs and adults) of six species of triatomines—*Triatoma maculata*, *T. dimidiata*, *R. pallescens*, *R. prolixus*, *Eratyrus* ssp. and *P. geniculatus*—from the field in Colombia, these contain a wide variety of bacteria, but some families of bacteria are more common in certain species [60]. Only *P. geniculatus* is not colonized by Nocardiaceae. In addition, some families of bacteria are only found in nymphs or are more common in nymphs than in adults, where bacterial diversity is lower than in nymphs.

### 4.3. Identification of Symbionts 

Investigations considering populations from laboratory colonies provide no information about the natural microbiota and symbionts, because triatomines acquire bacteria from other species reared in the same insectary (summarized by [150]) and probably also in the field. Therefore, the identification of *Rhodococcus rhodnii* in *Rhodnius ecuadoriensis* from the insectary [151] requires verification using field samples. In such specimens, we identified the symbiont of three species of triatomines. The symbiont of *T. infestans* is *Rhodococcus triatomae*, which was previously classified as a *Nocardia* sp. [152], but later re-named as *R. triatomae* [153]. In *Triatoma sordida* and *P. megistus*, an unnamed *Gordonia rubropertinctus*-like isolate and an unnamed *Rhodococcus equi*-like isolate, respectively, induce normal development [53]. These identifications are based on cultures using molecular biology methods. The long-known symbiont of *R. prolixus*, *Rhodococcus rhodnii*, also belongs to the Actinomycetales. Since only four species of symbionts have been identified and many other bacteria are present in the intestine, the symbiosis cannot yet be classified as a phylosymbiosis, defined as “microbiological community relationships that recapitulate the phylogeny of their hosts” [154], as in other Hemiptera [155]. Although the symbionts have only been identified in four triatomine species, the importance of Actinomycetales seems obvious. A special feature is the change in the growth pattern of Actinomycetales in the triatomine. Initially, individual cocci are present, which combine to form coccal chains that separate, or mycelia develop [150,156]. The mycelium-like appearance led to the name “Actinomycetes”. So far, Actinomycetales have sometimes not been found or not in all samples (see Section 4.2 and [127]). This seems to speak against the role of symbionts. However, the role of other bacteria and the origin of the blood must be taken into account (see Section 4.4 below). 

### 4.4. Functions of Symbionts/Microbiota 

Aposymbiotic triatomines develop a disease syndrome, particularly in the late nymphal instars. After feeding on guinea pigs, rabbits or humans, nymphal development is retarded, and adults rarely develop. In addition, tanning, digestion and excretion are disturbed, and the tracheal system is reduced (summarized by [157,158]). After feeding aposymbiotic nymphs of *R. prolixus* with blood supplemented with either B vitamins or the symbionts, development is much better than in aposymbiotic nymphs and no signs of aposymbiosis appear [156]. A function as a vitamin B supplier is therefore postulated. One argument against the vitamin B hypothesis is the feeding of various auxotrophic mutants of *R. rhodnii* that are unable to synthesize certain B-complex vitamins [159]. These nymphs develop normally. Furthermore, feeding aposymbiotic *T. infestans* throughout nymphal development with sterile defibrinated pig blood, or on mice and chickens, also avoids aposymbiosis effects (summarized by [157]). This presumably resulted in the development of a male *T. infestans* with disturbed tanning, which we captured in the field in Bolivia. 

In addition, some bacteria produce vitamin B_12_ [160]. *Bacillus megaterium*, which has the enzymes for the synthesis of this B vitamin [161], colonizes the intestines of field-derived *T. dimidiata* [162]. In *Drosophila*, the ability to produce a B vitamin differs between bacterial isolates of wild-caught flies [163]. According to a metagenome shotgun sequencing approach, various intestinal bacteria of triatomines possess the genes for the synthesis of the different B vitamins [164]. 

However, alternatives are possible. All four species of symbionts belong to Actinomycetales and to the mycolata taxon, in which the cell walls of bacteria contain mycolic acids [165,166]. Enzymes for the degradation of acylglycerols are also only present in the genome of *Rhodococcus* and not in other intestinal bacteria [149]. In addition, so far, the different forms of growth of this bacterium have also not been taken into account (see Section 4.3).

### 4.5. Development of Symbionts/Bacteria in Triatomines

The development of symbionts is strongly linked to the cardia, the first region of the midgut. In ants, this region acts as a micropore filter that develops after the establishment of symbiotic bacteria and then blocks the entry of other bacteria [167]. Stinkbugs of the family Acanthosomatidae possess midgut crypts that house the symbionts and are not connected to the midgut lumen [168]. In another hemipteran, the bean bug *Riptortus pedestris*, colonization of crypts in the posterior midgut by symbionts induces symbiont-mediated morphogenesis, with closure of the midgut preventing colonization of other bacteria, a widespread phenomenon in plant-feeding heteropterans [169,170,171]. In triatomines, the cardia has deep sac-like infoldings with narrow channels that open into the lumen. These are densely populated with bacteria, which are thereby protected from the complement factors in the ingested blood [172].

Focussing on the symbionts of *T. infestans* and *R. prolixus*, *R. triatomae* and *R. rhodnii*, respectively, we infected aposymbiotic first instar nymphs via a mixture of the respective symbiont and blood [152]. By covering the beakers with sterile aluminium foil and feeding each nymphal instar under sterile conditions via membranes, no contaminations with other bacteria occurred. When determining the numbers of colony-forming symbionts in the cardia, stomach, small intestine and rectum up to ten days after feeding of fifth instar nymphs, both symbiont–triatomine systems behave similarly: Numbers are lower one day after feeding than in the cardia and stomach of unfed nymphs, but then increase rapidly, in the cardia of *T. infestans* and *R. prolixus* up to five and seven days after feeding, respectively, and then remain at this level. In the stomach, the increase lasts up to seven/eight days after feeding and then also remains at this level. Ten days after feeding, 8 × 10^8^ colony-forming units/fifth instar stage are present in *R. prolixus* and 1.8 × 10^8^ in *T. infestans*, of which 95–99% are in the cardia and stomach. Passage into the small intestine results in a strong degradation in *T. infestans* and *R. prolixus* to 150,000 colony-forming units/small intestine and 90,000 colony-forming units/rectum. Most symbionts in the rectum are excreted within four hours after blood ingestion [152]. At seven days after the feeding of fifth instar nymphs of *R. prolixus*, approximately 60 × 10^8^ and 2 × 10^8^ colony-forming unit bacteria are present in the stomach and small intestine, respectively [173], but the symbionts are not counted separately.

Using the metagenome shotgun sequencing approach, the relative abundance of Corynebacteria, which includes *Rhodococcus*, is similar in the stomach and small intestine of *R. prolixus* three days after blood ingestion, but lower in the stomach two and seven days after feeding [149]. Since this contradicts the symbiont data (see above), many non-symbiotic Corynebacteria are probably present. The percentage of bacteria belonging to five bacterial orders also changes within seven days after feeding, but there is no evidence as to the cause of these changes.

Summarizing the data on the development of symbionts, after feeding, they develop strongly in the anterior regions of the midgut, and there are much fewer of them in the small intestine. The sequencing approach highlights the importance of separating the numbers of symbionts from those of other bacteria in the stomach and small intestine, especially considering different days after feeding. Changes in the abundance of Actinomycetales or *Rhodococcus* during ontogeny or in different regions of the intestine [148,174,175] require careful consideration [127].

### 4.6. Intestinal Bacteriolysis

When studying bacteriolysis in different regions of the intestine using turbidity assays and the lyophilized mutualistic symbiont *R. triatomae* as a substrate, no lysis occurs after incubation with homogenates of the stomach and small intestine of unfed *T. infestans* nymphs and those up to 50 days after feeding and buffers at pH 3 to pH 9 [176]. Using the Gram-negative bacterium *Escherichia coli* and the Gram-positive *Micrococcus luteus* as substrates, at pH 7, the extracts from the stomach exhibit higher activities than those from the small intestine. Such a difference in the activities of both intestinal regions is also evident after feeding in fifth instar nymphs of *R. prolixus* [177] and in the midgut glycosidases of this species, as well as in the prophenoloxidase activities of fifth instar nymphs of *T.* (*M.*) *pallidipennis* [178,179]. This corresponds to a higher expression level of genes encoding lysozymes and defensins in the stomach and cardia than in the small intestine of fifth instar nymphs of *Triatoma brasiliensis* and *T. infestans*, which is clearly visible in whole-mount in situ hybridization [180,181]. More differentiated expression is observed at seven days after feeding a mixture of blood and either the Gram-positive *Staphylococcus aureus* or the Gram-negative *Escherichia coli* to fifth instar nymphs of *R. prolixus* [177]: In the stomach, the transcripts of lysozyme A, B, defensin C and prolixicin predominate, while in the small intestine, the mRNAs are mainly coding for lysozyme B and prolixicin (the latter lyses Gram-negative *E. coli* [182]). While *S. aureus* induces an upregulation of the expression of *DefA* and *DefB* in the stomach tissue, only *DefC* is upregulated there after infection (p.i.) by *E. coli*. The expression rates of genes of prolixicin and three annexins are higher in the small intestine than in the stomach of fifth instar nymphs of *R. prolixus* one and seven days and one day after feeding, respectively [104,177]. Differential activations of the immunity by different bacteria also occur after feeding antibiotics to fourth instar nymphs of *R. prolixus*, followed by infections of fifth instar nymphs with the symbiont or the Gram-negative *Serratia marcescens* [183] and are common in other insects, e.g., aphids [184]. Since the synthesis of the antimicrobial compounds requires energy, this is reduced during starvation: In long-term-starved *T. infestans*, i.e., in adults fed 50 days before the dissection as fifth instar nymphs, bacteriolytic activity is present in the stomach but not detectable in the small intestine [176]. 

In zymographs of extracts of the stomach and small intestine of unfed *T. infestans* nymphs and those up to 50 days after feeding, using non-reducing sodium dodecyl sulfate-polyacrylamide gel electrophoresis and lyophilised *M. luteus* as an indicator, more bands of lysis develop when using extracts of the latter region. Lysis bands are mainly present at 15 to16 kDa and also at 36 and 40 kDa, but never in the molecular range < 14 kDa [176]. Using *T. infestans* saliva from fifth instar nymphs, unfed and seven days after feeding, nine lysis bands appear in the molecular range > 14 kDa, strong activities at 17.1 and 24.5 kDa, and also at 31.4 and 38.5 kDa. After a 24 h incubation in deionized water to demonstrate the bacteriolytic activity occurring under hypotonic conditions, additional lysis bands can be seen [185]. The appearance of lysis zones in the higher molecular range and after an incubation in deionised water indicates the presence of an undissociated complex of an antimicrobial peptide with another protein. Protein complexes are present in the saliva of triatomines [119,186]. In the stable fly *Stomoxys calcitrans,* intestinal defensins are bound intracellularly in SDS-stable complexes to a serine protease and released into the lumen of the gut [187]. Since bacteria in larvae of houseflies and *Drosophila melanogaster* are digested in a combination of low pH, aspartate proteases and lysozymes [188,189], such an effect is suspected in triatomines, where two cathepsin D genes show differential temporal expression after feeding [81]. Furthermore, a murein endopeptidase obtained through horizontal transfer from microorganisms to the genome of *R. prolixus* could be involved in symbiont digestion [190].

The strong development of the symbionts in the stomach and the decrease in the small intestine seem to contradict the high bacteriolytic activities in the stomach and the low activities in the small intestine. However, in these bacteriolytic assays, symbionts are not lysed by extracts from both regions [176], and the bacteriolytic compounds appear to regulate the non-symbiotic bacteria. The question of the factors that lyse the symbionts remains unresolved. These may be present in complexes of antimicrobial factors and digestive enzymes or the murein endopeptidase.

## 5. Interactions of Triatomines with *Trypanosoma cruzi*

Before summarizing the interactions, a critical comment on infection of triatomines is necessary. Only an infection via an infected mammalian host is optimal. Blood trypomastigotes should be used to study the initial development and response of the vector [191]. Any method of isolating blood trypomastigotes, metacyclic trypomastigotes and epimastigotes may affect the parasites (summarized by [192]). Isolation of blood trypomastigotes using DEAE-Sephacel ion exchange chromatography, high centrifugation forces and washing in salt solutions without proteins leads to the shedding of parts of the surface layer [192]. When using epimastigotes rather than blood trypomastigotes (e.g., [173]), only immune responses a few days later or after molting to the next instar could reflect responses following natural infections. Natural infections with epimastigotes occur after cannibalism, i.e., the ingestion of the contents of the stomach of the attacked triatomine [193], and after coprophagy, in which only small amounts of epimastigotes are ingested.

### 5.1. Effects of the Vector on Trypanosoma cruzi—Development of the Parasite in the Vector

The strain of *T. cruzi* and the species/strain of triatomine determine whether or not the parasite establishes in the vector and what role the triatomine plays in disease transmission (e.g., [47,53,127,194,195,196]). In triatomines in the field, the prevalence of *T. cruzi* increases from instar to instar, which is probably related to the number of feeding events [197]. High levels of human migration result in a strong mix of parasites and vectors. Upon introduction of a vector into new locations, these triatomines are susceptible to the local strains of *T. cruzi*, although often less susceptible than the local native vectors (e.g., [195,198,199,200]).

*T. cruzi* strains belonging to different discrete typing units are present in many geographical regions (e.g., [201,202]) (Table 1). There, mixed infections with *T. cruzi* occur in natural populations of triatomines, in which trypanosome strains behave differently [203]. After infection of *T. infestans* with different *T. cruzi* strains, some strains do not develop in all insects, and strains that develop lower numbers of trypanosomes have a lower percentage of metacyclic trypomastigotes/insect [204]. When studying the development of single and mixed infections of *T. cruzi* strains in *T. infestans* and *T. brasiliensis*, the population density of one strain is improved or affected by co-infection with another strain [205,206]. 

*T. cruzi* develops in the Malpighian tubules, midgut and rectum [34,41,207]. The four Malpighian tubules are colonized in only six out of ten *T. cruzi* (TcI)-infected *T. infestans*, the end of the Malpighian tubules, the ampullae, in eight out of ten nymphs, all of which have all stages of development but a very small number of trypanosomes [34]. In electron microscopy, the entire lumen of the ampullae is occupied by *T. cruzi*, but only after infection with one of two strains [207]. Epimastigotes are in contact with the microvilli of the ampullae [208]. Since colonization of the cardia has not yet been considered, the development in the stomach, small intestine and rectum is summarized below.

#### 5.1.1. Development of *Trypanosoma cruzi* in the Stomach

Since glucose is no longer available, *T. cruzi* must use amino acids and lipids for metabolism that come from the digestion of the membranes of blood cells and proteins in the blood plasma (see Section 3.2). This appears to be of lesser importance since the trypanosome has the ability to utilize carbohydrates and amino acids without drastic changes in its catabolic enzyme levels [209]. 

Only humoral, and not cellular, components of immunity of triatomines are active in the intestine. They and many other compounds, including anticoagulatory factors, are ingested with saliva, but also come from the stomach wall (see Section 3.3). The salivary factors present in *R. prolixus*, but not in *Rhodnius colombiensis*, that lyse epimastigotes of TcII, but not TcI [210], remain to be identified, but lysis of epimastigotes of *T. cruzi* in the stomach is only possible in established infections when the parasites have already colonized the small intestine and rectum [193]. In *T. infestans*, salivary trialysin lyses blood trypomastigotes [211], but only after trialysin is knocked out can its role in development be identified. Most of the factors in the stomach cannot be assigned to specific developmental steps of *T. cruzi* there (summarized by [53,127]). 

Initial development appears to depend heavily on the combination of parasite and vector. One day p.i. of *R. prolixus* with *T. cruzi* strain CL (TcVI), the number of blood trypomastigotes is greatly reduced, and four days after infection, no parasites are found in the stomach. Death of blood trypomastigotes is induced by extracts from the stomach of recently blood-fed *R. prolixus* and not by unfed specimens [212]. In another investigation of this system, a few hours p.i., blood trypomastigotes differentiate into amastigote-like forms, and agglutinations after one or two days of infection are followed by five days p.i. an elongation of the body, but no transformation into epimastigotes [213]. Most of the cell-culture-derived trypomastigotes of the clone Dm28c (TcI) are lysed in the stomach of *R. prolixus* during the first 24 h, but individual trypomastigotes are still present three weeks later [214]. A detailed analysis of the development of blood trypomastigotes of *T. cruzi* strain G (isolated after accidental laboratory infection with strain Y (TcII), hence the name change) in the stomach of *T. infestans* highlights the early development of spheromastigotes (which were first named with the German term Sphaeromastigote by this author) and aggregated dividing amastigotes [215]. Three and four days p.i. of nymphs of *T. infestans* with blood trypomastigotes of *T. cruzi* strain MR, the stomach contains a large number of pear-shaped forms and amastigotes, which aggregate and fuse, presumably as part of a genetic exchange. Within the first six days p.i., the “presence of epimastigotes could not be excluded with certainty” [216]. Only in the first ten days p.i of adults of *Triatoma phyllosoma pallidipennis* with a local Mexican *T. cruzi* strain are there amastigotes, but also epimastigotes, in the stomach [217].

Summarizing the development in the stomach, many blood trypomastigotes are lysed in some parasite–vector systems. In all systems, trypomastigotes transform into amastigotes and spheromastigotes, and epimastigotes rarely develop. 

#### 5.1.2. Development of *Trypanosoma cruzi* in the Small Intestine

In the small intestine of the vector, trypanosomes transform into epimastigotes. They multiply rapidly [71] because blood digestion supplies many more nutrients (see Section 3.2). The good development of *T. cruzi* indicates that its surface coat resists all digestive enzymes of the vector, proteases, lipases and enzymes for carbohydrate digestion. Epimastigotes are in contact with the perimicrovillar membranes and the microvillar border of the cells of the intestinal wall of the triatomines, but the perimicrovillar membranes are only transient, developing after blood ingestion and degenerating during starvation (see Section 3.2). In electron microscopy of nymphs of *T. infestans*, many flagellates can be seen near the perimicrovillar membranes, but not penetrating them [218,219]. In regions without membranes, flagella rarely interdigitate between microvilli [218]. In contrast to these figures, epimastigotes are thought to attach to the midgut via different compounds, but often under conditions that could induce attachment (summarized by [53,127]). Interaction with the perimicrovillar membranes is indicated after feeding antibodies against perimicrovillar membranes or phytochemicals or decapitation of *R. prolixus*, all of which strongly affect the development of the perimicrovillar membranes and reduce the populations of *T. cruzi* (summarized by [127]). However, this can also be due to impaired digestion. 

The rapid development of the population is shown in a system in which the vector and *T. cruzi* (TcI) originate from the same village in Chile: One week after the ingestion of 8000 to 10,000 blood trypomastigotes by each second instar nymph of *T. infestans*, the small intestine contains about 30,000 parasites/nymph [71]. After the feeding of subsequent nymphal stages at three, six and ten weeks p.i., the populations of flagellates increase in each successive nymphal stage. The small intestine of fifth instar nymphs is colonized with an average of up to 600,000 parasites/nymph [71]. Predominantly epimastigotes and various intermediate stages to spheromastigotes and trypomastigotes are present, but rarely metacyclic trypomastigotes with a kinetoplast in the subterminal position. 

While blood ingestion induces rapid population development of *T. cruzi* in each nymphal stage of this system, starvation affects the population density and the different stages. A slight reduction in population density is evident in the third, fourth and fifth nymphal instars before feeding after starvation periods of three or four weeks, and more round forms are present [71]. A more pronounced effect occurs in this system when fifth instar nymphs, originally infected in the first instar, are starved [220]. Twenty days after blood feeding of the fourth instar nymphs and two days after molting to the fifth instar, about 60,000 *T. cruzi* colonize the small intestine. Thirty days after feeding, 3000 flagellates are present, and after another 30 days, no more trypanosomes are found [220]. However, in previous determinations of starvation capacity in this parasite-vector system, the small intestine of all dead triatomines contained smaller populations of live trypanosomes [221]. In addition, the small intestine of dead nymphs contains remnants of digested blood. Therefore, probably, it is the loss of specific compounds, rather than too-low concentrations of digested hemoglobin, that results in the death of most of the *T. cruzi* population.

Summarizing the development of *T. cruzi* in the small intestine, blood ingestion supports the development of the population, and epimastigotes and spheromastigotes multiply rapidly. Rarely, final metacyclic trypomastigotes develop. Starvation affects the density of the population.

#### 5.1.3. Development of *Trypanosoma cruzi* in the Rectum

The development of *T. cruzi* in the rectum was also recently reviewed [53,127]. Summarizing the most important aspects, the population there is much larger than in the small intestine, three times larger for the *T. cruzi/T. infestans* system from Chile [71]. This seems to be due to the possibility of attachment, as about two thirds of the population are attached [71]. A small hydrophobic region on the flagellum is involved in attachment to the wax layer that covers the entire rectal cuticle [222,223]. Such an attachment in insects occurs in many genera of trypanosomatids [224]. The attachment region of the flagellum contains putative signaling domains [225] that may be involved in attachment. A surface mucin covering the whole flagellate [226] cannot be involved in the attachment of the flagellum. In established infections, the flagellum is enlarged at the attachment site [75]. A further indication of the strength of attachment is hemidesmosome-like material in the enlargement below the flagellar plasma membrane. Up to four layers of epimastigotes cover the cuticle, and the upper layers have an elongated flagellum whose tip is attached to the cuticle. When classifying the population density in different regions of the rectum, the rectal pads are particularly favored. In other insects, water and amino acids are strongly absorbed there (summarized by [227]), but the effects of *T. cruzi* on this process are unknown.

Attachment may provide an opportunity to interfere with the development of *T. cruzi*, as attachment promotes the development of infectious metacyclic trypomastigotes that occur only or mainly in the rectum. However, unattached epimastigotes also transform into metacyclic trypomastigotes. Trypomastigotes arise from different developmental stages, long and short epimastigotes, giant cells and spheromastigotes, but also from epimastigotes in which two different daughter cells arise through unequal cell division, an epimastigote and a metacyclic trypomastigote [71,72]. The proportion of metacyclic trypomastigotes in the total population depends on the duration of the infection, the triatomine species and the *T. cruzi* strains, and can be up to 50% in established infections (e.g., [228]; summarized by [53,127]). The development is influenced also by the ambient temperature (summarized by [229]). When examining the number of *T. cruzi* in the rectum of *T. pallidipennis* 5 to 60 days p.i. of fifth instar nymphs with two flagellate strains and a maintenance temperature of 20, 30 and 34 °C, the trypanosome populations increase steadily at both lower temperatures, but more strongly at 30 °C, while at 34 °C, an initial increase turns into a steady decline [230].

Blood ingestion and starvation affect not only the population in the small intestine but also that in the rectum. The urine induced and produced by feeding flushes out the population of *T. cruzi* in the rectal lumen and part of the population on the rectal wall, especially trypomastigotes, which cannot attach due to the short free flagellum and the surface coat. The population is reduced by around >50% (summarized by [72]). In addition, an interesting phenomenon occurs after feeding of fifth instar nymphs of *T. infestans* 22 days after molting, i.e., 40 days after blood feeding as fourth stage nymphs: metacyclogenesis is induced within four hours after blood ingestion, but only in epimastigotes [34]. The inducing factors are hemolymph proteins of about 17 kDa that pass into the urine [72,231]. They inform the epimastigotes: “Hurry up! You will be excreted and only metacyclic trypomastigotes will survive in the mammalian host.” There, the epimastigotes are lysed in a complement-mediated reaction [232]).

Experiments to induce metacyclogenesis in vitro consider only epimastigotes. They mimic induction by urine by incubating epimastigotes from the exponential growth phase for two hours in a saline solution—mainly sodium chloride—named triatomine artificial urine, which is then supplemented, e.g., with proline [233]. Within 24 h, the nutritional stress caused by incubation in the saline solution affects the morphology of the epimastigotes, resulting in a slimmer appearance and ultrastructural changes, e.g., in the kinetoplast [234]. Within 72 h after supplementation with proline, about 86% of the flagellates are metacyclic trypomastigotes [233]. In these assays, the epimastigotes require much more time than the four hours in the vector (see above), but the in vitro process allows detailed molecular biological and biochemical analyses during transformation. After induction of epimastigote-borne metacyclogenesis in vitro, significant different metabolic modifications are evident that are used by both forms to generate energy [235].

After long-term starvation, another feeding-induced phenomenon occurs [236]. Feeding *T. cruzi* (TcI)-infected fifth instar nymphs of *T. infestans* 42 days after the molt, i.e., 60 days after blood-feeding of fourth instar nymphs, results in a change in the proportions of the different stages. The most obvious forms at one day after feeding are about 10% “giant cells”, i.e., a multiple cell division stage. Two days later, their proportion increases to 30 to 50%, but none are found in later dissections [236]. They also develop in small numbers in the small intestine [215].

Starvation without subsequent feeding also affects the population density of *T. cruzi* (TcI) in *T. infestans*. About 300,000 flagellates colonize the rectum 20 and 30 days after feeding of the fourth instar nymphs [220]. After another 30 days, the population is reduced to 100,000 trypanosomes/rectum and 90 and 120 days after feeding to 1000. The reduction between 30 and 60 days after feeding may be at least partly related to the deposition of a drop of feces, but then almost all nymphs store rectal contents without defecating. Even 120 days after feeding, all recta remain colonized [220]. In scanning electron microscopy of fifth instar nymphs starved for 16 weeks, most rectal regions have attached flagellates. Four weeks later, only the rectal pads are still inhabited in all nymphs [237]. Starvation affects not only population density but also the frequency of a particular stage of *T. cruzi*, the spheromastigotes [220]. One day and 20 days after blood ingestion, 2% of the population are spheromastigotes and 1% are intermediate drop-like forms [220,236]. Forty days later, 30% of the rectal population are spheromastigotes and intermediate forms [236]. 

Does a nymph’s molt affect the rectal population [102]? Following the development of two strains of *T. cruzi in T. infestans* in weekly intervals, at one day after the molt of fourth instar nymphs, the population density appears to be similar or higher than in the previous week [71]. Presumably the epimastigotes detach from the old cuticle, and the rectal sphincter compresses this cuticle very strongly when it is torn out, giving the population an access to the rectal lumen and the new cuticle. This hypothesis requires counting the number of flagellates remaining on the rectal exuvia.

The time until the start of defecation determines the importance of a triatomine as a vector. This varies greatly between species (summarized by [53,127]). Species that excrete their feces after leaving the host are less important as vectors (e.g., [7,238]).

To summarize the development of *T. cruzi* in the rectum, the wax layer on the rectal cuticle enables epimastigote attachment and, presumably, thereby a better development of the population than in the small intestine. Epimastigotes and spheromastigotes multiply rapidly. Blood ingestion induces the production of urine, which flushes out the unattached population. Hemolymph proteins in the urine induce a rapid metacyclogenesis of epimastigotes, but not the other four pathways. After long-term starvation nymphs ingest blood, “giant cells” appear for a few days. During starvation, population densities progressively decrease favoring the development of spheromastigotes. 

### 5.2. Effects of Trypanosoma cruzi on Triatomines

#### 5.2.1. Effects of *Trypanosoma cruzi* on Nymphs and Adults of Triatomines 

Several reviews focus on the effects of *T. cruzi* infection on triatomines (e.g., [41,53,78,83,127,196,227,239,240,241]). According to these reviews, *T. cruzi* either has a weak pathogenicity or is subpathogenic, i.e., pathological effects only develop when triatomines are exposed to adverse conditions, e.g., lack of symbiont supply, starvation, suboptimal temperatures and suboptimal blood sources, all of which affect the development of even uninfected triatomines [55,56,242,243,244]. Infection with symbionts is particularly crucial in laboratory experiments, but this information is rarely included in the publications.

Although the mean duration of starvation resistance in fourth and fifth instar nymphs of *T. infestans* is statistically significantly shortened by up to 17% after first instar infection compared to uninfected nymphs [221], this shortening is unlikely to have a significant impact on population development as the overall natural mortality rate (egg to adult) is estimated at 86% [245]. The starvation ability of infected *T.* (*M.*) *pallidipennis* is also affected, but not that of *Mepraia spinolai* from the field [179,246]. However, the nutritional status of infected specimens is lower than that of uninfected *M. spinolai* [247]. Three days p.i. with epimastigotes, the lipid metabolism of male *R. prolixus* is impaired [248]. In nymphs of this species, some *T. cruzi* strains significantly reduce survival and retard the development, while after infections with other strains, development resembles that of uninfected nymphs [249]. Studies of adult longevity and fertility also show conflicting results, but some strains of *T. cruzi* appear to affect the vector even under optimal conditions ([250]; summarized by [53,127]). The high population densities of *T. cruzi* in the small intestine and rectum are likely to affect the metabolites there. This is confirmed by intensive analyses, in which the composition of metabolites in the intestinal tract of infected *R. prolixus* differs from that of uninfected specimens [251]. According to an analysis of free aminoacids and those in peptides, *T. cruzi* influences the composition of peptides in rectal contents of *T. infestans*, presumably through proteases in the surface coat of the epimastigotes [252].

#### 5.2.2. Effects of *Trypanosoma cruzi* on the Behavior of Triatomines

Triatomines exhibit a wide range of different behaviors (summarized by [59,253]), and most of these have not been compared between uninfected and *T. cruzi*-infected triatomines. The abundance of some sensilla types on the antennae of *Triatoma dimidiata* originating from domestic and sylvatic populations differs between uninfected and infected insects, e.g., more mechanoreceptors are present in infected ones and more chemoreceptors in the infected sylvatic population [254]. This does not affect host selection, as infected and uninfected *M. spinolai* choose the same hosts [255]. The effects of *T. cruzi* infection on triatomine orientation to the host were recently summarized [53,127]. Often, infected nymphs respond more quickly to human odors or approach the host more quickly than uninfected nymphs [256]. Within the group of infected *M. spinolai*, a higher nutritional status correlates with a shorter period of time to approach [247], i.e., increasing duration of starvation does not induce a faster response, but there is an optimal time for feeding (see Section 3.1). Since *Triatoma rubrovaria* infected with *T*. *cruzi* (TcVI) ingest more blood than uninfected nymphs [257], *T. cruzi* and the vector probably compete for components in the blood [221]. Then, a more advanced state of starvation in infected triatomines leads to behavioral responses that appear somewhat later in uninfected nymphs. It is questionable whether an earlier approach of nymphs, requiring only one complete engorgement per instar, increases the risk of transmission. Starvation also reduces the locomotion of infected nymphs and increases the dispersal ability of infected female *T. dimidiata* by reducing their weight (summarized by [53,127]). Furthermore, both sexes of infected *T. infestans* show higher levels of negative geotaxis and higher levels of aggregation [258]. A final aspect is that the aggregation of infected *T. pallidipennis* is impaired due to a lower concentration of attractive compounds in dry feces [259].

#### 5.2.3. Effects of *Trypanosoma cruzi* on Digestion and Immunity

Effects on digestive enzymes have rarely been investigated. Nine days p.i. with cell-culture derived trypomastigotes of Colombiana strain, intestinal expression of genes encoding a main digestive enzyme, cathepsin B, is similar in infected and uninfected adults of *Rhodnius neglectus*, but expression of cathepsin D is significantly reduced by infection [120]. However, compared to uninfected fifth instar nymphs of *R. prolixus*, one day and three days p.i. with epimastigotes, the cathepsin D activity is higher in the small intestine [89]. 

*T. cruzi*-infections in the triatomine intestine enhance the immune response of hemocytes, fat body and other organs (summarized by [53,127,240,260]). Hemocytes appear to be of no importance to *T. cruzi*, which does not invade the hemocoel [127]. However, in the early stages of infection and before the passage to the small intestine, hemocytes, fat body and the cells of the small intestine produce, in a systemic reaction, low-molecular mass compounds, including nitric oxide, which are involved in antibacterial responses [97]. A transport into the intestine is possible. A short-term effect at one day p.i. with blood trypomastigotes of *T. cruzi* strain Y (TcII) is evident in the stomach of *T. infestans* [261,262]. Whereas the expression of some genes of antimicrobial peptides is upregulated, others remain unchanged (summarized by [53,127]). These responses are induced by the surface coat of blood trypomastigotes of of *T. cruzi* (TcI): Up to five days after ingestion of blood mixed with both blood trypomastigotes or their shed surface coat, antibacterial activity is significantly increased in the small intestine of *T. infestans* nymphs [263]. After the feeding of trypomastigotes without the surface coat or epimastigotes, no reaction occurs ([263]; details in [53,127]). Using the whole intestine, the expression of a lysozyme gene is increased in *R. prolixus* at seven and 14 days p.i. with blood trypomastigotes of the Colombian HA strain [264]. At 20 days p.i. with epimastigotes of *T. cruzi* (TcII), the expression rate of the gene encoding a defensin in the small intestine of *T. brasiliensis* is much higher than in uninfected nymphs [265], and immune responses differ in such infections with different *T. cruzi* strains (summarized by [266]). Investigating the intestinal transcriptome of *Rhodnius neglectus* at two and nine days p.i. with cell-culture derived trypomastigotes of the Colombiana strain of *T. cruzi* (TcI), the gene expression of defensin C is significantly reduced [120]. Since recombinant defensin, based on a sequence from *T.* (*M.*) *pallidipennis*, kills *T. cruzi* in vitro [267], this interaction requires further investigation. The expression of the genes encoding nitric oxide synthase and phenoloxidase is increased after infection in several *T. cruzi*-vector systems (summarized by [127,266]). 

Summarizing the effects of *T. cruzi* on nymphs and adult triatomines, the intensity of effects depends heavily on the respective parasite–vector system. The effects on populations in the field can hardly be estimated. The quicker approach of nymphs to the host is irrelevant for the risk of transmission, because usually they require only one complete engorgement per instar. While there are no relevant effects on digestion, the immune response of triatomines is induced by the surface coat components of ingested blood trypomastigotes. However, this only seems to be relevant for the interactions with the intestinal microbiota of the triatomine (see chapter 6. below).

## 6. Interaction of *Trypanosoma cruzi* and the Microbiota of Triatomines 

### 6.1. Effects of the Microbiota on Trypanosoma cruzi and Competition of Both

There are few investigations of direct effects of the microbiota on *T. cruzi*. Lysis of epimastigotes by *Serratia marcescens* (summarized by [127]) appears to depend on the strain of the bacterium [268], and this bacterium is also present in triatomines infected with *T. cruzi* (e.g., [269]). For 20 years, genetically-transformed symbionts that produce a lepidopteran antibacterial peptide, dsRNA, an enzyme disrupting surface glycoconjugates, or an antibody fragment that all kill *T. cruzi* in the gut have been proposed for use in control campaigns [270,271,272,273,274], summarized by [275]. However, the release of transformed bacteria into the field is still controversial. 

Considering indirect effects in the interactions between *T. cruzi* and the intestinal microbiota, there is only little evidence of competition between flagellates and bacteria for resources through a different trypanosome–triatomine system: In *T. infestans* fed with blood supplemented with several B-group vitamins, the homoxenous trypanosomatid *B. triatomae* develops better than in nymphs fed unsupplemented blood, indicating competition of the trypanosome and the vector for these compounds [276].

### 6.2. Indirect Effects of Trypanosoma cruzi via Inducing Vector Immunity

These interactions are described in several reviews (e.g., [53,100,127,277,278,279]). After infection, blood trypomastigotes induce an upregulation of the synthesis of triatomine antimicrobial compounds, which suppress the growth of bacteria and promote the growth of flagellates. *T. cruzi* infections also suppress the development of intestinal fungi [124]. Less bacteria and more trypanosomes also develop after feeding antibiotics to act against the bacteria or an inhibitor of nitric oxide synthase [280,281]. In *R. prolixus*, it coincides with an increase in mRNA levels of defensin C and prolixicin, as well as the activity of phenoloxidase and antibacterial activity and a decrease of the synthesis of precursors of nitric oxide [173,280]. According to a metagenome shotgun sequencing approach, *T. cruzi* significantly reduces the abundance of bacteria after feeding in the stomach and the small intestine and changes the bacterial species composition [149] (see Section 4.2 and Section 4.5).

After knockdown of antimicrobial peptides, the synthesis of which is induced by *T*. *cruzi*, or suppression of immune responses by physalin B, a plant-derived immune modulator of *R. prolixus*, more intestinal bacteria and a lower number of flagellates develop compared to the control group [117,118,282]. However, these negative correlations of the population densities of bacteria and *T. cruzi* cannot be generalized. Silencing an antibacterial rhamnose-binding lectin of *R. prolixus*, whose transcript abundance is higher in the stomach than in the small intestine, and in both regions is higher at 12 than at 6 days after feeding, increases the transcript levels of bacteria but does not affect the population of *T. cruzi*, which developed after infection with epimastigotes [283]. 

In triatomines captured in natural habitats, the bacterial species composition of the microbiota is similar or more diverse in infected and uninfected specimens [143,144,145,175,269,284]. The latter could be due to long-term infections, since after feeding a mixture of different bacteria or fungi and blood, high numbers of the microorganisms develop only in nymphs long-term infected with *T. cruzi* TcI, but not in uninfected controls, indicating immune suppression in the intestine [285]. However, *T. cruzi* infections seem to suppress specific bacteria and support other bacteria. Infected *T. infestans* synthesize TiAP, an antimicrobial compound from the anterior midgut, and recombinant TiAP has bacteriostatic effects against Gram-negative *E. coli* and not against the Gram-positive *Micrococcus luteus* [118]. In comparison to uninfected specimens, in the feces of 59 *T. infestans* from the field infected with *T. cruzi* (TcI), species of five genera of bacteria are overrepresented—*Sporosarcina*, *Nestenrenkonia*, *Alkalibacterium*, *Peptoniphilus*, *Marinilactibacillus* and an unclassified genus of Porphyromonadaceae—and four are underrepresented—*Bosea*, *Mesorhizobium*, *Dietzia* and *Cupriavidus* [286]. In *T. dimidiata* from Colombia, infections with *T. cruzi* TcI lead to better development of Kineosporiaceae but suppression of Brevibacteriaceae, Dermabacteriaceae and Enterobacteriaceae [60]. The microbiota in natural populations of *T. sanguisuga* even shows a significant association of specific bacteria and discrete typing units. Bacillales are overrepresented in infections with TcI, Aeromonadales with TcIV and Burkhodeliales and Enterobacteriales with TcII/V [146]. In the hindgut of 74 specimens of *Triatoma gerstaeckeri* and *T. sanguisuga* from the field, the bacteria are not associated with the discrete typing units TcI and TcIV, but more Enterobacterales and *Petrimonas* develop in infected specimens [147]. 

Taking these inductions of vector immunity by *T. cruzi* together, the complexity of the short-term response by which *T. cruzi* induces the synthesis of antibacterial compounds that kill certain bacteria that appear to compete with the flagellate becomes apparent. The growth of other bacteria is supported, sometimes depending on the discrete typing unit of the flagellate. During long-term infections, *T. cruzi* suppresses intestinal immunity, which may be the reason for an overrepresentation of some bacteria in triatomines from the field. 

### 6.3. Interactions of Trypanosoma cruzi with Wolbachia sp. and Symbionts

*Wolbachia* should be given special attention because it supplies bed bugs with vitamin B and has genes for the synthesis of all B vitamins [287,288]. In bed bugs, elimination of the intracellular *Wolbachia* symbiont, which is specifically localized in bacteriomes, induces a disease syndrome similar to the effects seen in aposymbiotic triatomines [287,288,289]. In field-caught triatomines, all organs and all or many specimens are infected with *Wolbachia* [144,286,289]. However, in some regions, this bacterium is not present, and in Panama, it infects *R. pallescens* in one province and not another [144], and no effects on the triatomines are reported. Before its classification as a mutualistic triatomine symbiont, the procedures carried out on bed bugs must be repeated with triatomines. Since co-infections occur [144], an effect of *Wolbachia* on *T. cruzi* is unlikely.

Interactions between the symbiont and *T. cruzi* have rarely been investigated. At seven days p.i., more trypanosomes are present in *R. prolixus*, which has an established infection with the symbiont, than in aposymbiotic nymphs, but at 21 and 35 days p.i., the opposite is evident [290]. The percentages of the different stages of *T. cruzi* are similar. Six and 24 h after feeding blood with or without *R. rhodnii*, but with cell culture-derived trypomastigotes, to germ-free first instar nymphs, the population density of *T. cruzi* is similar in both groups [214]. After feeding the symbionts *R. rhodnii* and *R. triatomae* to axenic first instar nymphs of their respective triatomines, *R. prolixus* and *T. infestans*, and axenic maintenance and sterile feedings on hens, subsequent infection with blood trypomastigotes of *T. cruzi* (TcI) in the fifth instar and dissections up to ten days p.i., the population densities of the respective symbionts in the cardia, stomach, small intestine and rectum are similar to those in uninfected nymphs [152]. Knockdown of the immune component rpRelish, which appears to control defensin A expression, increases the population of *R. rhodnii* in the stomach, small intestine and rectum of *R. prolixus* (in that descending order), but levels of *T. cruzi* remain unchanged at seven and 14 days p.i. [291]. However, at seven days p.i. of *R. prolixus* with epimastigotes, the population density of *R. rhodnii* is significantly reduced [173]. These discrepancies could be due to the mode of infection or strain peculiarities of *T. cruzi*.

Summarizing the co-infections of triatomines with *T. cruzi* and *Wolbachia*, no interactions appear to occur. This is also evident in co-infections with the symbiont.

## 7. Suggestions for Future Research

Investigations should reflect the natural scenario and not be induced by laboratory conditions. Uniform protocols should be used [42]. With regard to *T. cruzi*, only the blood trypomastigote is the optimal stage to infect triatomines. *T. cruzi* strains should be isolated from triatomine populations captured in natural habitats and classified. Isolates belonging to one or more discrete typing units should be cloned, short-term cultured in vitro and then used to infect nymphs of the respective triatomine species. Cloning is necessary to avoid infections in which subpopulations of different clones develop at different times p.i. (see chapter 5.1). Aliquots of feces and urine should be mixed with an antifreezing compound and stored at −80 °C for infecting experimental mammals in subsequent experiments with the same material. Following the development in these mammals at different days p.i., perhaps higher percentages of slender and stout blood trypomastigotes occur [292], and the development in the triatomine can be studied, optimally in the progeny of triatomines from which *T. cruzi* has been isolated. This makes it possible to determine in how many natural systems the ingested blood trypomastigotes are killed in the stomach of nymphs and whether this differs between slender and stout trypomastigotes. The use of new techniques, such as fluorescence microscopy, bioluminescence imaging and qPCR will allow us to follow up the fate of parts of the population. Thereby, the great diversity of the *T*. *cruzi* strains becomes visible, and perhaps also some general peculiarities.

Identification of the molecular components within the hydrophobic binding zone of the flagellum of epimastigotes in the rectum of the vector [223] could lead to an approach to interfering with the development of *T. cruzi*. However, this will be difficult because even the detailed elaborations of the transcriptome in the context of in vitro metacyclogenesis studies [293,294] do not yet offer an approach to interrupt transmission.

Taking the vectors into account, other species of Triatominae should be investigated. Fewer than 10 of approximately 150 species of triatomines are included in interaction studies. Using other species could reveal new interactions. When focusing on specific issues, the role of the cardia should be taken into account. Is it “just” protecting symbionts in the deep infoldings or does it have a part in the recognition of foreign organisms? The latter is indicated by the relatively high expression levels of the lysozyme genes [180]. An open question is the fate of glucose from the ingested blood, as glucose is suggested to induce transformation into epimastigotes and influence flagellum length (summarized by [127]). Another important open question is the digestive process in the stomach. According to proteomics before and after blood ingestion, more proteolysis proteins are synthesized there than in the small intestine [295]. Are these proteins just precursors of enzymes and active in the small intestine or is hemoglobin already digested in the stomach without any noticeable change in color? The conditions in the rectum also require further investigation, including small molecules [296]. We know that oxygen concentrations are very low, but we can no longer carry out detailed studies. 

Including the flagellates, the role of trypanolytic factors in the stomach remains unclear (see Section 5.1.1). Are these salivary compounds also active in the small intestine? Epimastigotes dominate there and secrete cyclophilin, which inactivates the vector-derived salivary trialysin [297]. Investigations of interactions of flagellate and vector should begin with freshly established triatomine stocks from the field. The population used to isolate *T. cruzi* is optimal. For many insect species, only a small proportion of a group reproduces in the insectarium. This was also my experience with *T. infestans*. Therefore, high numbers should be caught in the field. In the insectary, these should be divided into three to four groups with at least 30 specimens each. The groups should be reared separately for a few years, then mixed for one generation and divided again. This allows genetic diversity to be restored. We mixed three stocks 12 years after the colonies were founded, and this is probably why the development of the *T. cruzi* strains that were temporarily stored at −80 °C was very similar over the >20-year period of our studies.

Much more research is needed on symbionts since symbionts allow triatomines to develop into adults and produce offspring. Therefore, adults from the field would provide the opportunity to identify the symbionts. Only adults from the field provide information about symbiont infection rates in a triatomine population. Adults with blood residue in their stomach, in particular, are ideal because the symbionts multiply in the stomach a few days after blood ingestion and are only present in small numbers in the rectum. Since details of the identification procedure of symbionts have recently been published [126], I will only describe this briefly and add possible modifications. The stomach and rectal contents of fifth instar nymphs or adults from the field should be plated on agar. All colonies should be stored. Although *T. cruzi* also develops on agar plates, the flagellate can be excluded using light microscopy. Since only a few symbionts are passed to the rectum, this makes identification easier. Actinomycetales—recognizable by the color change of the colonies to pink/red after a few days—should be separated and cultivated. After disinfection of the triatomine eggs, germ-free first instar nymphs (ideally the offspring of triatomines caught in the same location as the animals used for isolation of bacteria) should be infected via a mixture of blood and the cultures of Actinomycetales from a single colony on the agar plates. An alternative is to feed with uninfected blood and place the culture at the bottom of the beaker with the nymphs. Not all Actinomycetales establish in triatomines [150], but one criterion for a symbiont is the establishment. The main difficulties are sterile feeding, preferably with blood from sheep or cows, and axenic maintenance to the adult stage. A maintenance in beakers covered with sterile aluminum foil does not eliminate the risk of contamination; it only reduces it. Therefore, the identity of the bacterium used for infection should be verified in the feces of older nymphs. It would be optimal to keep them in a sterile incubator and provide them with sterile air, but this has never been tried. It may be possible to shorten and simplify the process by using third instar nymphs from the colonies and feeding them with antibiotics. This greatly reduces the microbiota [183]. After infection with Actinomycetales, only the symbionts prevent the development of the disease syndrome associated with aposymbiosis (see Section 4.4). Groups that develop to adults have received the symbiont. To underline a clear statement: Only bacteria that are fed to germ-free nymphs and enable good development are symbionts. However, these groups should be fed on the blood of sheep, cows or humans, but not on pig blood or on mice and chickens (see Section 4.4). Bacteria that enable good nymphal development should be identified using molecular biological methods. The function of the symbionts can then be clarified. This open question requires a biochemical determination of the different B vitamins in symbiont-infected nymphs and, in a molecular biology approach, the determination of the presence of the genes required for the synthesis of the different B vitamins. If B vitamins are not present, the supporting compound should be identified.

There are many other open questions about the microbiota, e.g., determining the period of coprophagy (see Section 4.1). Since the symbionts are not affected by the high antibacterial activity in the stomach, the following questions arise: what compounds kill them in the small intestine, which has much lower antibacterial activity? Do the compounds come from complexes of antibacterial factors and digestive enzymes? (See Section 4.6).

In addition, co-infections with bacteria occurring in the rectum that do not belong to the Actinomycetales should be used, e.g., to analyse interactions with the symbionts. The interactions between the different species of bacteria have not yet been investigated in triatomines. At least bacteria from the *Bacillus subtilis* group produce antimicrobial compounds [298] and could thereby change the composion of the microbiota in triatomines. Such a bacterium is present in both midgut regions of field-derived *T*. (*M*.) *pallidipennis* [284]. After infection of nymphs with a mixture of blood and small numbers of the different bacteria, including the symbiont, the establishment and population development of the different bacteria should be followed during nymphal development, including time points before and after feeding. Since not all triatomines in the field have actinomycetes, other bacteria appear to act symbiotically, although the blood donor hosts should be identified to rule out an effect of this (see chapter 4.4). Fungi also still need to be examined. Are they just occasional infections or are they digested and provide the triatomines with specific compounds? This occurs in other yeast-insect interactions (e.g., [299]).

Furthermore, it still needs to be investigated how *T. cruzi* achieves the effects on the microbiota. After an infection of germ-free nymphs with symbionts and a Gram-positive or Gram-negative bacterium and an infection with *T. cruzi* in the next instar, the number of symbionts and other bacteria should be determined in the stomach and small intestine, and the expression rate of the genes of the various antimicrobial compounds should also be determined. The use of *T. cruzi* strains of different typing units allows identification of the surface compounds of *T. cruzi* that initiate the immune response. 

## 8. Conclusions 

It will be difficult to find new approaches to prevent the transmission of Chagas disease by triatomines, although new methodologies allow us to discover new aspects of the interactions of the three main players, the trypanosomes, the triatomines and the microbiota. With the help of optimal systems, the question of the pathogenic effect of *T. cruzi* on the vector can be clarified. We should avoid generalizing results obtained in a single *T. cruzi*/triatomine system because the populations of *T. cruzi*, the triatomines, and the microbiota are too diverse. Many questions about the interactions of *T. cruzi*, triatomines, and the microbiota require investigations, particularly the molecules present in the intestinal tract that determine which *T. cruzi* strain and bacterium establish themselves in the vector. The use of metagenomic, proteomic and metabolomic data will provide new insights into these trypanosome–triatomine–microbiota systems.

## Figures and Tables

**Figure 1 microorganisms-12-00855-f001:**
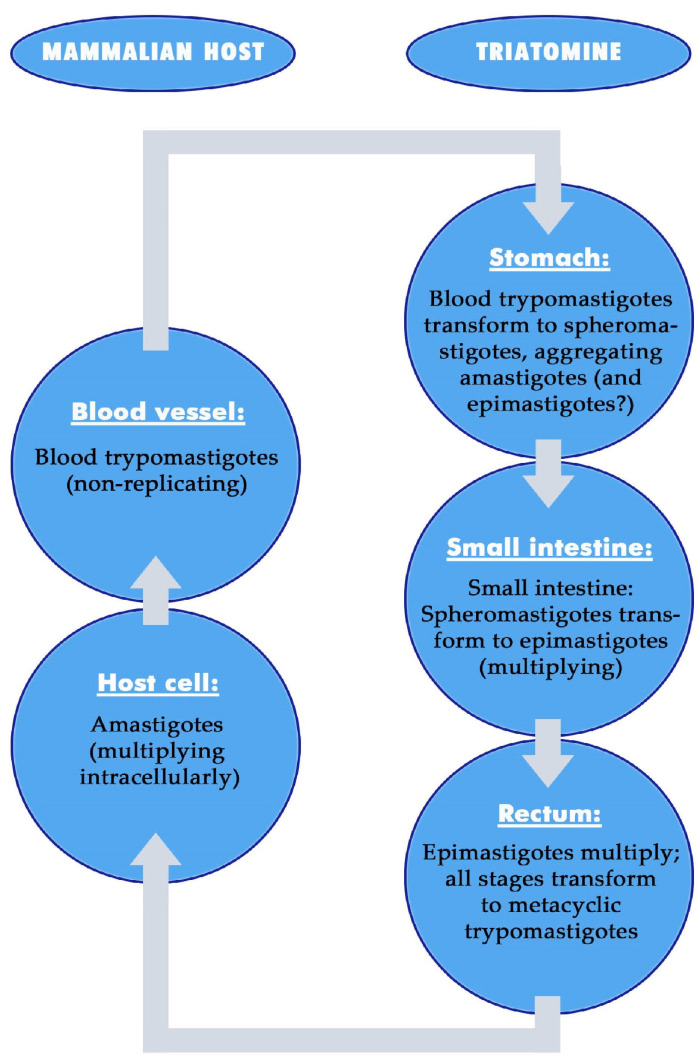
Developmental cycle of *Trypanosoma cruzi* in mammals and triatomines.

**Table 1 microorganisms-12-00855-t001:** Distribution of discrete typing units (DTUs) of *T. cruzi* and important species of triatomines in different countries of Latin America.

Country	DTUs [43]						Triatomines [44,45]
	TcI	TcII	TcIII	TcIV	TcV	TcVI	
Argentina	+	+	+		+	+	*T. infestans*, *T. guasayana, T. patagonica, T. sordida*
Belize	+			+			*T. dimidiata, R. pallescens*
Bolivia	+	+	+	+	+	+	*T. infestans*, *T. sordida*
Brazil	+	+	+	+	+	+	*T. infestans*, *T. brasiliensis*, *T. sordida*, *P. geniculatus, P. megistus*
Chile	+	+	+		+	+	*T. infestans*
Colombia	+	+	+	+	+	+	*T. dimidiata, R. prolixus, P. geniculatus*
Costa Rica	+						*T. dimidiata,* *P. geniculatus*
Ecuador	+	+		+	+	+	*T. dimidiata*, *T. carrioni*, *R. prolixus, P. rufotuberculatus, P. geniculatus*
El Salvador	+				+	+	*T. dimidiata*
French Guiana	+						*R. prolixus*
Guyana		+					*R. prolixus*
Guatemala	+						*T. dimidiata*
Honduras	+	+		+		+	*T. dimidiata*
Mexico	+	+	+	+	+	+	*T. dimidiata, T. protracta, T. barberi, M. longipennis*, *M. pallidipennis*
Nicaragua	+						*T. dimidiata* *, P. geniculatus*
Panama	+	+			+	+	*T. dimidiata, R. pallescens* *, P. geniculatus*
Paraguay	+	+	+		+	+	*T. infestans*, *T. sordida, P. geniculatus*
Peru	+		+	+	+		*T. infestans*, *P. chinai, P. geniculatus, R. ecuadoriensis*
Suriname		+					*R. prolixus,* *P. geniculatus*
Uruguay		+			+	+	*T. infestans*, *T. sordida, P. geniculatus, P. megistus*
USA	+	+		+	+	+	*T. sanguisuga, T. lecticularia, T. protracta*
Venezuela	+	+	+	+	+	+	*T. dimidiata, T. maculata*, *R. prolixus, P. rufotuberculatus, P. geniculatus*

Abbreviations of genera: *M*.: *Meccus*; *P*.: *Panstrongylus*; *R*.: *Rhodnius*; *T*.: *Triatoma.*

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
