# Peer review of "Interaction of Trypanosoma cruzi, Triatomines and the Microbiota of the Vectors—A Review"

_microorganisms, 2024, doi:10.3390/microorganisms12050855_

Round 1
Reviewer 1 Report
Comments and Suggestions for Authors
In this review, Günter A. Schaub presents a clear and complete summary of the state of knowledge about the interaction of Trypanosoma cruzi, the causing agent of Chagas disease, their insect vectors, the triatomines, and the microbiota of the vectors. The author focuses on what is known about the biology of the triatomine and its microbiota, the different stages of development of T. cruzi infection inside the vector and the impact of T. cruzi infection on the insect vector and its microbiota. The author also mentions the many gaps in the current knowledge and suggests questions to be investigated in the future. This reviewer believes that this paper is a mandatory reading for all researchers working in the field of parasite-vector biology. Overall, the work is well written and presented and covers up-to-date research on the topic. However, there are minor suggestions that can help to improve the presentation of the work:
1. Several mentions of Trypanosoma cruzi should be in italics: Title of chapter 2, Title of the sections 5.1.1, 5.1.2, 5.1.3, 5.2.1, 5.2.2, 5.2.3.
2. Missing comma and hyphen in the sentence: “During blood ingestion, the fastest fluid-secreting cells in the animal kingdom, the upper Malpighian tubuli cells, begin to produce urine.” (lines 159-160)
3. A few sentences have words with different font sizes:
- “…sodium dodecyl sulfate-polyacrylamide gel electrophoresis…” (lines 456-457)
- “…enzymes for carbohydrate digestion.” (line 562)
- “When examining the number…” (line 631)
- “Hemocytes appear to be of no importance to…” (lines 759-760)
4. The sentence below (lines 545-548) is a little confusing. It is unclear how one strain was isolated after an accidental laboratory infection with another strain. Also, it is not clear who is the person who named the spheromastigotes. Since the original reference is in German, this reviewer recommends rephrasing the sentence to clarify these points.
“A detailed analysis of the development of blood trypomastigotes of T. cruzi strain G (isolated after accidental laboratory infection with strain Y) in the stomach of T. infestans, highlights the early development of spheromastigotes (which were first named by her) and aggregated dividing amastigotes [211].”
5. The mention of TiAP (lines 835-836) also needs clarification. This reviewer suggests the following addition:
“Infected T. infestans synthesize TiAP, an antimicrobial product from the anterior midgut, and recombinant TiAP has bacteriostatic effects against Gram-negative E. coli and not against the Gram-positive Micrococcus luteus [115].”
6. In the References section, the reference numbers are duplicated.
Author Response
Reviewer 1
In this review, Günter A. Schaub presents a clear and complete summary of the state of knowledge about the interaction of Trypanosoma cruzi, the causing agent of Chagas disease, their insect vectors, the triatomines, and the microbiota of the vectors. The author focuses on what is known about the biology of the triatomine and its microbiota, the different stages of development of T. cruzi infection inside the vector and the impact of T. cruzi infection on the insect vector and its microbiota. The author also mentions the many gaps in the current knowledge and suggests questions to be investigated in the future. This reviewer believes that this paper is a mandatory reading for all researchers working in the field of parasite-vector biology. Overall, the work is well written and presented and covers up-to-date research on the topic. However, there are minor suggestions that can help to improve the presentation of the work:
I thank reviewer 1 very much for these initial comments.
- Several mentions of Trypanosoma cruzishould be in italics: Title of chapter 2, Title of the sections 5.1.1, 5.1.2, 5.1.3, 5.2.1, 5.2.2, 5.2.3.
Changed. These errors were caused by editorial changes as the submitted version had all section headings in italics. I don´t know how to write these names in headings written in italics (chapter 5.1, 5.2, 6.1, 6.2 and 6.4 (incl. Wolbachia).
- Missing comma and hyphen in the sentence: “During blood ingestion,the fastest fluid-secreting cells in the animal kingdom, the upper Malpighian tubuli cells, begin to produce urine.” (lines 174-175).
Changed.
- A few sentences have words with different font sizes:
- “…sodium dodecyl sulfate-polyacrylamide gel electrophoresis…” (lines 474-475)
- “…enzymes for carbohydrate digestion.” (line 583)
- “When examining the number…” (line 635)
- “Hemocytes appear to be of no importance to…” (lines 781-782)
Changed. These errors arose during the creation of the journal version. I apologize that it was my fault for not reading this version carefully enough.
- The sentence below (lines 545-548) is a little confusing. It is unclear how one strain was isolated after an accidental laboratory infection with another strain. Also, it is not clear who is the person who named the spheromastigotes. Since the original reference is in German, this reviewer recommends rephrasing the sentence to clarify these points.
“A detailed analysis of the development of blood trypomastigotes of T. cruzi strain G (isolated after accidental laboratory infection with strain Y) in the stomach of T. infestans, highlights the early development of spheromastigotes (which were first named by her) and aggregated dividing amastigotes [211].”
Changed to (lines 565-569): “A detailed analysis of the development of blood trypomastigotes of T. cruzi strain G (isolated after accidental laboratory infection with strain Y, hence name change) in the stomach of T. infestans, highlights the early development of spheromastigotes (which were first named with the German term Sphaeromastigote by this author) and aggregated dividing amastigotes [211].”
- The mention of TiAP (lines 835-836) also needs clarification. This reviewer suggests the following addition:
“Infected T. infestans synthesize TiAP, an antimicrobial product from the anterior midgut, and recombinant TiAP has bacteriostatic effects against Gram-negative E. coli and not against the Gram-positive Micrococcus luteus [115].”
Changed. Now lines 857-860.
- In the References section, the reference numbers are duplicated.
Changed. These errors arose during the creation of the journal version. I apologize that it was my fault for not reading this version carefully enough. However, such errors should not be introduced by the editorial team.
Reviewer 2 Report
Comments and Suggestions for Authors
This Review paper summarises the most recent advances on the research about gut interactions occurring in trypanosome vectoring hemiptera, which is a certainly relevant topic due to the medical significance of Chagas disease. The general organization of the manuscript is well conducted and allow a clear understanding of the covered issue; some suggestion to improve the text are listed below:
Introduction: a scheme or figure describing the whole trypanosome life cycle involving the vector and the human host would be useful to better delineate the following sections describing microbial interactions in the insect gut. For example, it would be immediately clear that the trypanosome exploits the vector as a crucial host for its development, and for this reason there is a stable and consistent localization in the rectum allowing us to take advantage of microbial partners to counteract the parasite.
Section Vectors: again I suggest adding a visual support (e.g. a map) to immediately resume the geographical distribution of different vector species, maybe also in combination with trypanosome strains. Adding a table could be an alternative option.
Section Identification of symbionts: since most of the mentioned symbionts are cultivalble, whereas the majority of gut symbionts from other hemipterans are not, it should be specified when identification was obtained by cultivation and when by sequencing (if so).
Section Development of symbionts/bacteria in triatomines, Lines 386-389. Here I suggest adding also the following reference: Oishi, S., Moriyama, M., Koga, R., and Fukatsu, T. (2019). Zool Lett. 5, 16. doi: 10.1186/s40851-019-0134-2, to show that the closure of midgut crypts is widespread in plant feeding heteropterans.
Section Interactions of Trypanosoma cruzi with Wolbachia sp. and symbionts, lines 865-866. Please specify if there is any evidence of the localization of Wolbachia in triatominae, i.e. in bacteriocytes like in bed bugs or in reproductive tissues like in most of other insects.
An additional suggestion about this section is to include a short paragraph describing the attempts that have been made to genetically manipulate symbiotic bacteria to kill trypanosome; this approach may also be shortly discussed in the conclusions.
Author Response
Reviewer 2
This Review paper summarises the most recent advances on the research about gut interactions occurring in trypanosome vectoring hemiptera, which is a certainly relevant topic due to the medical significance of Chagas disease. The general organization of the manuscript is well conducted and allow a clear understanding of the covered issue; some suggestion to improve the text are listed below:
I thank reviewer 2 very much for these initial comments.
Introduction: a scheme or figure describing the whole trypanosome life cycle involving the vector and the human host would be useful to better delineate the following sections describing microbial interactions in the insect gut. For example, it would be immediately clear that the trypanosome exploits the vector as a crucial host for its development, and for this reason there is a stable and consistent localization in the rectum allowing us to take advantage of microbial partners to counteract the parasite.
I included a scheme.
Section Vectors: Again I suggest adding a visual support (e.g. a map) to immediately resume the geographical distribution of different vector species, maybe also in combination with trypanosome strains. Adding a table could be an alternative option.
I included a table.
Section Identification of symbionts: since most of the mentioned symbionts are cultivalble, whereas the majority of gut symbionts from other hemipterans are not, it should be specified when identification was obtained by cultivation and when by sequencing (if so).
In lines 314-315, I added that all new publications that used molecular biology methods. I also added in lines 357-358 that these identifications of mutualistic symbionts are based on cultures using molecular biology methods.
Section Development of symbionts/bacteria in triatomines, Lines 386-389. Here I suggest adding also the following reference: Oishi, S., Moriyama, M., Koga, R., and Fukatsu, T. (2019). Zool Lett. 5, 16. doi: 10.1186/s40851-019-0134-2, to show that the closure of midgut crypts is widespread in plant feeding heteropterans.
In line 406, I included this reference and in the text: “a widespread phenomenon in plant-feeding heteropterans”.
Section Interactions of Trypanosoma cruzi with Wolbachia sp. and symbionts, lines 865-866. Please specify if there is any evidence of the localization of Wolbachia in triatominae, i.e. in bacteriocytes like in bed bugs or in reproductive tissues like in most of other insects.
Wolbachia is present in all organs. I added this in line 885.
An additional suggestion about this section is to include a short paragraph describing the attempts that have been made to genetically manipulate symbiotic bacteria to kill trypanosome; this approach may also be shortly discussed in the conclusions.
This had already been included, now in lines 816-823, and I added a summarizing review. I would not like to include it into the Conclusions. The release of transformed bacteria into the field is still controversial. One of my PhD students, Stefan Eichler, had worked in the lab of Ben Beard on this topic (Beard CB, Dotson EM, Pennington PM, Eichler S, Cordon-Rosales C, Durvasula RV. Bacterial symbiosis and paratransgenic control of vector-borne Chagas disease. Int J Parasitol. 2001 May 1;31(5-6):621-7. doi: 10.1016/s0020-7519(01)00165-5). I assume that the most recent attempts will not receive approval for release into the field either.
Reviewer 3 Report
Comments and Suggestions for Authors
I would like to congratulate the author for this review article. This review manuscript is wonderfully well written in context of showing vivid information and relatedness to the topic. Every headings and sub-headings are elaborately described with proper citations. Overall, it is a well documented and balanced article to show the interaction and whole pathogenesis of Trypanosoma Cruzi and relatedness to the vectors and their microbial contributions.
Comments on the Quality of English LanguageThe manuscript is very well written. However, in my opinion, there are some trivial issues regarding sentence construction, font size and spelling error that can be corrected as well.
Line 66-68: The sentence construction can be rephrased better while mentioning the first discovery of trypanosome infection in the patient by Dr. Chagas.
Line 456- 457: The font size of the part of the sentence is different. Is it to highlight the text or just an unintended editing problem? 'polyacrylamid' should be 'polyacrylamide'.
Line 562: Font size is different here while mentioning the effect of the enzyme for carbohydrate digestion.
Line 759: Font size is different here while mentioning that hemocytes are having no effect to T. Cruzi invading the hemocoel.
Line 928: Author may consider to re-construct this line as assertive sentence.
Line 980: Author may consider to re-construct this line as "the following questions may arise" instead of "the question arises" as there are multiple questions mentioned afterwards.
Line 995: Author may consider to re-construct this line as assertive sentence.
Author Response
I would like to congratulate the author for this review article. This review manuscript is wonderfully well written in context of showing vivid information and relatedness to the topic. Every headings and sub-headings are elaborately described with proper citations. Overall, it is a well documented and balanced article to show the interaction and whole pathogenesis of Trypanosoma cruzi and relatedness to the vectors and their microbial contributions.
I also thank reviewer 3 very much for these initial comments.
Comments on the Quality of English Language
The manuscript is very well written. However, in my opinion, there are some trivial issues regarding sentence construction, font size and spelling error that can be corrected as well.
Sometimes I also had a strange feeling. However, the manuscript had been corrected by a native English senior scientist, Prof. Norman Ratcliffe. As already answered to Reviewer 1, the font size errors arose during the creation of the journal version. I apologize that it was my fault for not reading this version carefully enough.
Line 66-68: The sentence construction can be rephrased better while mentioning the first discovery of trypanosome infection in the patient by Dr. Chagas.
I replaced “trypanosomes” by “T. cruzi” (line 66).
Line 456- 457: The font size of the part of the sentence is different. Is it to highlight the text or just an unintended editing problem? 'polyacrylamid' should be 'polyacrylamide'.
As already mentioned, the font size errors arose during the creation of the journal version. I corrected to 'polyacrylamide'. (Now line 475)
Line 562: Font size is different here while mentioning the effect of the enzyme for carbohydrate digestion.
See above. Now line 583.
Line 759: Font size is different here while mentioning that hemocytes are having no effect to T. Cruzi invading the hemocoel.
See above. Now lines 781-782.
Line 928: Author may consider to re-construct this line as assertive sentence.
I would like to highlight this as a question. Now line 951.
Line 980: Author may consider to re-construct this line as "the following questions may arise" instead of "the question arises" as there are multiple questions mentioned afterwards.
Changed, but not including “may”. Now line 1003.
Line 995: Author may consider to re-construct this line as assertive sentence.
Again, I would like to highlight this as a question. Now lines 1019-1020.
